# Advances in the Diagnosis and Treatment of Rotavirus Infections: Narrative Review

**DOI:** 10.3390/ijms26189175

**Published:** 2025-09-19

**Authors:** Karolina Pawłuszkiewicz, Emilia Kucharczyk, Matylda Korgiel, Tomasz Busłowicz, Anita Faltus, Natalia Kucharczyk, Emil Paluch

**Affiliations:** 1Faculty of Medicine, Wroclaw Medical University, Wybrzeże L. Pasteura 1, 50-367 Wrocław, Poland; karolina.pawluszkiewicz@student.umw.edu.pl (K.P.); emilia.kucharczyk@student.umw.edu.pl (E.K.); matylda.korgiel@student.umw.edu.pl (M.K.); tomasz.buslowicz@student.umw.edu.pl (T.B.); anita.faltus@student.umw.edu.pl (A.F.); 2Health Care Complex in Oława, ul. Baczyńskiego 1, 55-200 Oława, Poland; 3Department of Microbiology, Faculty of Medicine, Wroclaw Medical University, St. T. Chałubińskiego 4, 50-376 Wroclaw, Poland

**Keywords:** rotavirus infections, viral diarrhea, diagnostic methods, dPCR, PAGE, new enzyme immunoassay

## Abstract

Rotavirus remains one of the leading causes of severe gastroenteritis, particularly among infants and young children, despite the introduction of effective vaccines. Although the global burden of rotavirus-associated morbidity and mortality has decreased in recent years, significant challenges remain regarding accurate diagnosis, optimal clinical management, and equitable access to preventive measures. The aim of this narrative review is to provide a comprehensive synthesis of recent advances in the diagnosis and treatment of rotavirus infections. Particular emphasis is placed on post-guideline research emerging after the publication of the Evidence-Based Guidelines from the European Society for Pediatric Gastroenterology, Hepatology, and Nutrition (ESPGHAN) and the European Society for Pediatric Infectious Diseases (ESPID), offering updated perspectives on therapeutic strategies and clinical practices. In addition, this review discusses the expanding role of molecular diagnostic methods, which offer enhanced sensitivity and specificity in the detection of rotavirus, and evaluates novel antiviral agents under investigation. By integrating and analyzing the most relevant research published within the past decade, we aim to delineate key progress, identify persistent gaps in knowledge, and propose future directions for research and clinical application.

## 1. Introduction

Rotavirus (RV) is the primary cause of severe gastroenteritis in children under five years of age [1]. First identified in 1973 from duodenal biopsies and fecal specimens of individuals with acute gastroenteritis (AGE), RV remains a significant global health concern [2]. Before the advent of RV vaccines, RV infections were a leading cause of severe pediatric AGE, contributing to approximately 500,000 deaths among children each year. They were implicated in 30% to 50% of all hospital admissions for AGE in children under the age of five, underscoring their substantial global burden prior to widespread immunization efforts [3]. In high-income countries with established vaccination programs, the incidence of RV infection has declined significantly. However, in low- and middle-income countries where vaccine coverage is limited, RV remains a leading cause of life-threatening diarrhea in young children [4].

RVs are non-enveloped, double-stranded RNA (dsRNA) viruses classified by the current International Committee on Taxonomy of Viruses (ICTV) system within the genus *Rotavirus* and family *Sedoreoviridae* [5]. Based on antibody reactivity and VP6 protein sequence identity, the recognized rotavirus species include A, B, C, D, E, F, G, H, I, and J [2], among which species A is responsible for over 90% of infections, while species B and C are only sporadically detected in humans [6,7]. They are characterized by a structure consisting of three concentric capsids, enclosing a genome composed of 11 dsRNA segments. All segments encode 12 viral proteins: 6 structural (VP1–VP7) and 6 non-structural (NSP1–NSP6) [6]. Figure 1 presents the structure of rotavirus and the associated functions of its structural viral proteins (VPs) [6,8].

Group A rotaviruses (RVA) are classified into distinct genotypes—42 for VP7 (G types) and 58 for VP4 (P types)—enabling the use of the binomial genotyping system for confirmatory vaccines design and characterization [7].

RVs are highly contagious and environmentally stable, facilitating their broad dissemination. They are transmitted primarily by the fecal–oral route, through both direct person-to-person contact and indirect exposure to contaminated surfaces or objects. Shedding of the virus in the feces can begin 2 days before the onset of symptoms and persist for several days after their resolution. Very high concentrations of the virus are present in the feces of infected individuals, making RV one of the most contagious pathogens causing diarrhea [1,4]. Clinical RV infection may involve the shedding of more than 10^12^ virus particles per gram of feces. The virus remains viable for months, contributing to its widespread transmission. In temperate climates, RV illness is most common during colder months, likely due to increased transmission within households and communities [2,9]. Hospital-acquired infections (HAIs) are common, and the virus can become endemic in neonatal hospital nurseries. While most human RV infections are not zoonotic—despite RVAs being widespread in young mammals—humans can occasionally be infected with reassortant strains originating from both human and animal sources. The role of such strains in driving new epidemics, either locally or globally, is still unclear [10].

Clinical manifestations typically include profuse diarrhea, vomiting, fever, and general malaise [1,11]. Moreover, emerging research suggests that RV may play a contributory role in a range of extraintestinal manifestations, including central nervous system complications, autoimmune conditions, biliary atresia, and certain respiratory tract disorders [8,12]. The combination of diarrhea and vomiting frequently leads to severe dehydration and reduced oral intake, often necessitating hospitalization and, if untreated, potentially leading to death [1,2].

RV antigens can be detected in stool samples using enzyme-linked immunosorbent assay (ELISA) and immunochromatographic (ICA) methods, while reverse transcription polymerase chain reaction (RT-PCR) assays are employed for the detection of viral RNA [1]. RT-PCR offers higher sensitivity and allows for viral genotyping. Confirmation of RV infection relies on stool sample testing through validated diagnostic assays [2,4]. Currently, RT-PCR is considered the reference standard for RV diagnosis. However, its widespread clinical application is limited by high costs, extended turnaround times, and the need for specialized laboratory infrastructure [13]. Antigen-based tests typically detect RV virions for up to seven days following symptom onset, whereas RT-PCR can identify viral RNA for extended periods—ranging from 4 to 57 days. Notably, RT-PCR may yield positive results in approximately 29% of asymptomatic children under one year of age. Given the simplicity, rapid turnaround, and low cost, ICA-based antigen detection is often recommended for routine use, especially in resource-limited settings [1,13]. However, false-positive antigen results are relatively common in recently vaccinated children, particularly outside hospital settings. These false positives can be misinterpreted as vaccine failure, potentially undermining public confidence in vaccine efficacy. As such, diagnostic testing plays a critical role not only in clinical management but also in disease surveillance and assessment of vaccine performance. To accurately distinguish vaccine-derived strains from wild-type viruses, routine RT-PCR testing is recommended in infants [2,4].

This review offers a comprehensive overview of the diagnostic and therapeutic approaches to RV AGE and is an in-depth analysis of recent advancements in RV diagnostics and treatment strategies. Importantly, this review emphasizes the role of molecular diagnostics in RV detection, showcasing advanced methods that markedly enhance sensitivity, specificity, and turnaround time. We also assess recent therapeutic studies published since the ESPGHAN/ESPID Evidence-Based Guidelines, highlighting the most up-to-date clinical evidence. Additionally, we examine the antiviral potential of investigational agents—ranging from natural compounds and host-directed molecules to pharmacological inhibitors—targeting distinct stages of the RV life cycle. This review synthesizes recent studies to map advances, outline remaining challenges, and suggest future directions in RV research. It serves two purposes—acquainting laboratory and diagnostic specialists with current molecular methods for RV detection and providing clinicians with evidence-based guidance on therapeutic options, ranging from established standards to emerging antivirals and supportive care.

## 2. Diagnostic Methods for RV Infections

### 2.1. Clinical Diagnosis

The clinical presentation of RV AGE varies widely, ranging from asymptomatic or mild, self-limiting watery diarrhea to severe illness characterized by frequent, profuse diarrhea accompanied by vomiting and fever. In severe cases, this may lead to dehydration, hypovolemic shock, electrolyte disturbances, and potentially death. Following an incubation period of 1 to 3 days, RV infection typically begins abruptly and presents with a range of clinical manifestations [14]. The most common clinical features include fever, diarrhea, and vomiting, which may occur individually or in combination. In one study of hospitalized children presenting with diarrhea, vomiting, and unexplained fever, the most frequent symptom combination was concurrent diarrhea, vomiting, and fever (63%). Overall, 97% of cases involved diarrhea and/or vomiting, with 91% presenting with diarrhea either alone or alongside other symptoms [14,15]. Approximately 30–40% of affected children develop moderate fever, with temperatures exceeding 39 °C. Vomiting generally subsides within one to two days, whereas gastrointestinal symptoms typically resolve within three to seven days [16].

Assessment of diarrhea-related dehydration relies on several validated clinical scoring systems, each characterized by distinct criteria, strengths, and limitations. As shown in Table 1, the Vesikari Scoring System (VSS), the Clark scale, and the Clinical Dehydration Scale (CDS) emphasize clinical features such as frequency of diarrhea, vomiting episodes, and degree of dehydration, whereas the Dehydration: Assessing Kids Accurately (DHAKA) and the Centre for Infectious Disease Research in Zambia (CIDRZ) scales were designed for rapid application in low-resource or high-burden settings [17,18,19,20,21,22,23].

These distinctions have important implications for both clinical practice and research. For example, the VSS, which is frequently employed in clinical trials for RV vaccines, offers a comprehensive disease severity score based on symptom duration and treatment intensity; however, because it relies on meticulous clinical monitoring, it is less suitable for standard care [24,25]. In contrast, despite their moderate accuracy, the CDS and Gorelick scales are more convenient to use in hectic pediatric settings [26]. The Gorelick scales, which include both 10- and 4-point versions, emphasize practical bedside evaluation of dehydration severity. The WHO and Gorelick scales, however, have limited diagnostic utility according to systematic evidence, particularly in children with acute gastroenteritis [27].

The DHAKA and CIDRZ scales were developed and validated in low-resource clinical settings to improve dehydration assessment during acute pediatric diarrheal illness. Strong inter-rater reliability and positive likelihood ratios, along with empirical data from a cohort of patients with acute diarrhea in Dhaka, Bangladesh, supported the diagnostic accuracy of the DHAKA Dehydration Score and Decision Tree in under-resourced environments [20]. The CIDRZ diarrhea severity scoring tool was developed and evaluated through outpatient surveillance in Lusaka, Zambia. It performed similarly to the DHAKA score and exhibited high reliability and internal consistency [21]. When taken as a whole, these context-specific tools demonstrate the usefulness of locally developed, validated scales designed for environments with few clinical resources.

Therefore, Table 1 highlights the variety of dehydration assessment instruments as well as the crucial significance of selecting tools based on context. While frontline clinicians in resource-limited settings may benefit from simpler tools like the DHAKA scale, which maintain a balance between practicality and reasonable accuracy, providers in tertiary hospital or research settings may prefer the comprehensive and trial-aligned Vesikari or Clark scales.

Among them, the Vesikari scale [17] and the Clark scale [18] are widely adopted tools, initially developed to gauge the severity of acute AGE, particularly in the context of evaluating RV vaccine efficacy. Figure 2 presents a comparison of both scales scores [17,18].

The Vesikari scale has become one of the most widely adopted tools for assessing pediatric gastroenteritis severity, supported by a 2011 manual that standardized its use and facilitated score calculation [28,29,30]. Its modified version—the Modified Vesikari Severity Score (MVSS)—has since been validated in large multicenter cohorts. Table 2 summarizes the key differences between the original VSS and the MVSS, providing a quick reference for clinical and research use. Wikswo et al. demonstrated in more than 14,000 US children that the MVSS strongly correlated with illness duration, hospitalization, and both school and parental work absenteeism [31]. Similarly, Schnadower et al., in a prospective study across five US emergency departments, confirmed the scale’s reliability, validity, and generalizability, with scores correlating with dehydration, hospitalization, and absenteeism [32], highlighting its utility even in acute care settings.

When compared with the Clark scale, the Vesikari system appears more sensitive. A London study of 200 children with RV-induced gastroenteritis found that 57% were classified as severe by the Vesikari scale compared with only 1.5% by the Clark scale (*p* < 0.001), with 24% of cases labeled mild by the Clark scale identified as severe by the Vesikari scale [24]. This difference is likely due to the absence of a dehydration component in the Clark scale, which reduces its sensitivity. Consistent results from India also showed that the Clark scale underestimates clinically severe cases requiring intravenous rehydration [24,34]. Consequently, the Vesikari scale is a more effective endpoint measure in clinical trials evaluating severe RV gastroenteritis and vaccine efficacy than the Clark scale [24,35,36].

In contrast, the CDS [19], the DHAKA score [20], the more recently developed CIDRZ scale by St. Jean et al. [21], Gorelick scales (4-points and 10 points) [22], and the WHO dehydration scale [23] focus specifically on assessing the degree of dehydration regardless of its etiology; their scores are compared in Figure 3 [19,20,21,22,23].

These tools play an important role in both clinical decision-making and research settings, especially in low-resource environments. However, as highlighted by the European Society for Pediatric Gastroenterology, Hepatology, and Nutrition/ European Society for Pediatric Infectious Diseases (ESPGHAN/ESPID) Evidence-Based Guidelines Working Group, currently available scoring systems may offer clinical utility, yet no single tool has achieved universal acceptance or standardization in routine assessment [37]. St. Jean et al. summarizes the precision–recall area under the curve (PR-AUC) values for the Vesikari, Clark, DHAKA, and CIDRZ scales (the latter developed by the authors) collecting data from March 2019 to July 2019 in Luska, Zambia, as shown in Table 3 [21]. These results indicate that dehydration-focused tools, such as the DHAKA and CIDRZ scores outperform the Vesikari and Clark scales in predicting clinical outcomes in young children, particularly in resource-limited settings.

Hoxha et al. [38], in a prospective study, evaluated the diagnostic performance of three clinical dehydration assessment tools—the WHO scale, the Gorelick scales, and the CDS—in children aged 1 month to 5 years with acute diarrhea, using ROC curve analysis against the gold standard of percent weight change with rehydration, as presented in Table 4 [38]. In this study cohort, the CDS was the least effective tool for predicting dehydration, performing notably worse than both the WHO dehydration scale and the Gorelick scales.

Despite continuous advancements in clinical practice, a universally endorsed and validated scale for reliably quantifying the severity of diarrheal disease and dehydration across diverse geographic and clinical settings is still lacking.

Hospital admission for AGE should be considered in the presence of profound dehydration exceeding 9% of body weight, neurologic impairment such as lethargy, stupor, or seizures, intractable or bilious vomiting, failure of oral rehydration therapy, clinical suspicion of surgical pathology, or when safe outpatient care and adequate home monitoring cannot be ensured [37].

### 2.2. Laboratory Diagnosis

Electron microscopy (EM) was the first method used for the diagnosis of RVs. Since the 1980s, ELISA tests have been commonly used, providing satisfactory results compared to EM. Recently, molecular techniques such as RT-PCR have replaced other diagnostic tests, and RT-PCR is now considered the gold-standard method for RV detection; however, these techniques require expensive equipment and advanced technical skills [39].

Traditional diagnostic methods are lengthy to operate and technically demanding [40]. Culturing the virus in cell lines is challenging and is not employed for diagnostic purposes. Additionally, serological tests lack clinical relevance because the majority of the population already possesses RV-specific antibodies [41]. EM is highly specific for RV detection and has a sensitivity comparable to some enzyme immunoassays (EIA), but lower than that of PCR. However, its use in routine testing of large numbers of stool samples is limited due to its time-consuming nature. Additionally, EM requires expensive equipment and highly trained personnel, and it cannot distinguish between different groups of RVs [42].

To detect viruses via EM, viral titers of approximately 10^6^/mL are necessary; such levels of viral shedding are typically observed during the first 48 h of infection. The sensitivity of detection can be enhanced 10- to 100-fold with immune EM, which also improves specificity. However, the availability of specific reagents remains relatively limited [43].

The latex agglutination test is rapid (less than 15 min), easy to use, and does not require sophisticated technology to perform, which makes it a useful tool for detecting epidemic outbreaks [44].

ELISA is one of the most widely used tests due to its high sensitivity, specificity, and relatively low cost, with the ability to process up to 96 samples simultaneously. However, it is time-consuming because it requires multiple washing steps to remove non-specifically bound reactants [40,45]. The ELISA test is used to detect the RV VP6 antigen—part of the inner capsid—in stool samples, which can confirm RV infection. It is a method routinely used for certain groups, e.g., Group C RV in the United Kingdom [46].

ICA is characterized by a short detection time [43]. Commercially available kits, such as the SD Rota/Adeno Rapid test, are clinically useful for the rapid diagnosis of RV infections. A study conducted in Korea on the aforementioned test suggests that clinicians should be aware of the potential for false-positive results, especially when the ICA band intensity is weak [47]. Studies in Gabon on another rapid diagnostic test (RDT), the SD BIOLINE Rota/Adeno Ag RDT, have shown that it has lower diagnostic accuracy compared to RT-qPCR. Additionally, it was found that while this test is a good method for diagnosing infectious intestinal disease, it is not suitable for screening asymptomatic populations for the purpose of preventing RVA transmission [40].

Polyacrylamide Gel Electrophoresis (PAGE) is a well-established, ultrasensitive method for detecting RV infections, particularly in resource-limited settings [48,49]. This technique involves the separation of dsRNA segments of the virus by size, followed by visualization through silver staining or other staining methods [50]. Fecal samples from suspected cases are processed for RNA extraction, after which the RNA is subjected to PAGE and visualized using silver staining or equivalent techniques [45,46]. Silver staining enhances the detection of dsRNA bands, enabling identification of low viral loads with high sensitivity and specificity [51,52]. A study conducted in Thailand reported that PAGE detected RV in 96.7% of 1304 stool samples, demonstrating performance comparable to that of ELISA [52]. Additionally, PAGE facilitates the differentiation of RV strains by electropherotype, thereby contributing to insights into genetic diversity [53]. Compared to other molecular diagnostic methods, PAGE is relatively inexpensive and requires minimal specialized equipment, making it particularly useful for surveillance and outbreak investigations in low-resource settings [52,53].

A meta-analysis of 12 studies [54], involving a total of 4407 children with RV-induced AGE, indicates the high sensitivity and specificity of ICA for RV detection, albeit with lower sensitivity than RT-PCR. The reduced specificity in detecting RV can be compensated for by serial testing [1,54]. Subgroup analysis of children aged ≤5 years showed slightly lower sensitivity compared to the overall combined sensitivity, which can also be mitigated by serial testing [54].

RT-PCR is among the most sensitive and specific tests (90–95%) and allows for the detection of small quantities of the viral genome [55]. Although it requires special equipment, specialized technicians, and is time-consuming, it is widely used in research studies [56].

### 2.3. Radiological Diagnosis

Several alternative methods for assessing hydration status have also been explored, although these typically require specialized equipment. These include ultrasound-based measurements of the inferior vena cava (IVC) diameter, inferior-vena-cava-to-aorta-diameter ratio (IVC/Ao), aorta-to-IVC ratio, and IVC inspiratory collapse [57].

The initial research to evaluate pediatric dehydration with ultrasonographic assessment was conducted by Kosiak et al., who also initiated the trend of utilizing the IVC/Ao ratio as a routine diagnostic indicator [58]. Octavius et al. conducted a meta-analysis of five studies in total involving 461 children, evaluating the diagnostic accuracy of the IVC/Ao ratio for clinically significant dehydration. The overall sensitivity of these studies was 86% (95% confidence interval [CI]: 79–91%), and the specificity was 73% (95% CI: 59–84%), with a comparatively high diagnostic performance of the tested method [59]. In a prospective observational study, El Amrousy et al. validated the accuracy of IVC/Ao ratio for the detection of significant dehydration in 200 infants. By the gold standard of percent weight change, only 134 infants were indeed significantly dehydrated. Using a prehydration IVC/Ao ratio cut-off point of <0.75, the method had a sensitivity of 82%, specificity of 91%, and overall accuracy of 87% [60]. This difference underscores the need for more high-quality multicenter trials to establish ultrasonographic criteria in a standardized fashion and validate their applicability across diverse clinical settings. Most importantly, the research of El Amrousy et al. illustrates poor performance of traditional clinical judgment (sensitivity, 70%; specificity, 63%), which was significantly less accurate in diagnosis than objective ultrasound evaluation. Such observations justify the possible usefulness of incorporating sonographic assessment into routine assessment procedures for pediatric dehydration, especially in settings where quick, non-invasive, and accurate decision-making is crucial. Radiological evaluation is necessary if neurological symptoms such as seizures or encephalopathy occur. Magnetic resonance imaging (MRI) can reveal abnormalities, such as hyperintense lesions resulting from edema, when present [9,61].

## 3. Innovative Diagnosis Methods for RV Infections

Recently, several new diagnostic methods have been developed, which may allow for the detection of RV infection with less sample material and with greater sensitivity.

### 3.1. Enzyme Immunoassay (EIA)

The EIA is a rapid method developed for the detection of RV antigens in fecal specimens. It is designed to provide a quick and reliable diagnosis of RV AGE, particularly in clinical settings where timely results are crucial [62,63]. The assay delivers results in approximately 10 min, offering a significant advantage over traditional methods that may take several hours. In a study involving 137 patients, the new EIA exhibited 89.2% sensitivity and 90.0% specificity before resolution. After applying a blocking reagent, sensitivity increased to 100%, and specificity remained high at 98.9% [63]. When compared to the Pathfinder RV assay and EM, the new EIA demonstrated competitive performance. In a study with 100 fresh stool samples, the sensitivity and specificity were 95% and 90%, respectively, for new EIA, compared to 84% and 98% for Pathfinder, and 63% and 100% for EM [64]. Although TESTPACK established the utility of rapid immunoassays for rotavirus diagnosis in clinical and outbreak settings, most published evaluations are dated; recent performance assessments of contemporary rapid rotavirus antigen tests emphasize molecular confirmation (e.g., RT-PCR) as the reference standard [64,65].

Recent advances in EIA for RV detection aim to increase sensitivity, particularly in specimens with low viral shedding, such as those from vaccinated infants. In a study by Wang et al., researchers developed a new biotin–avidin sandwich EIA using a monoclonal antibody (1D4) targeting the VP6 antigen of RVAs [65]. The study evaluated 128 fecal samples from infants who had received the Rotarix vaccine in Mexico, comparing the new assay with two commercial EIA kits, using RT-PCR as the reference standard. The new assay detected the RV antigen in 36.7% of samples, nearly double the detection rate of the commercial kits—which detected 16.4% and 18.0%—without cross-reactivity to other enteric viruses. This demonstrates improved sensitivity at lower antigen levels, while retaining specificity, making it a promising tool for diagnostics in vaccine trials and surveillance, where existing EIAs may miss low-level shedding [65].

Recombinant-antigen- and peptide-based EIAs that use anti-VP6 capture and multi-epitope detector antibodies have similarly shown high diagnostic sensitivity and specificity across multiple host species, supporting VP6 as a robust target for RVAs antigen detection [66]. More recently developed assays have expanded beyond manual microplate formats to include VP7-specific potency EIAs for vaccine-related work and automated fluorescent immunoassays (e.g., AFIAS) that offer a rapid turnaround, strong concordance with ELISA/PCR, and workflow advantages for high-throughput laboratories [67,68]. Independent evaluations of contemporary rapid fecal antigen kits report variable sensitivity—often lower than molecular methods—but generally high specificity. This underscores that, while next-generation EIAs narrow the gap with molecular tests for many surveillance applications, RT-PCR remains the reference standard when maximal analytical sensitivity or genotyping is required [67,68].

### 3.2. Digital PCR (dPCR)

Digital PCR (dPCR) is an advanced molecular technique that offers high sensitivity and precision in detecting low-abundance nucleic acid targets. In the context of RV infections, dPCR has been explored as a diagnostic tool, particularly for environmental surveillance and wastewater-based epidemiology [69]. dPCR has been utilized to monitor RV presence in untreated sewage, providing insights into community-level viral loads and trends over time. This approach aids in assessing the prevalence of RV infections in populations, including asymptomatic individuals [70,71].

dPCR enables the detection of rare mutations and alleles within the RV genome, such as single nucleotide variants (SNVs). This capability is crucial for understanding genetic diversity and evolution of the virus, which can inform vaccine development and epidemiological studies [69]. dPCR can detect low-abundance targets, making it suitable for identifying infections in asymptomatic individuals or in environmental samples with low viral loads. The partitioning of samples in dPCR allows for accurate quantification of viral genomes, providing detailed insights into viral loads. dPCR can simultaneously detect multiple pathogens, facilitating comprehensive diagnostic panels [69,70].

dPCR represents a promising advancement in the molecular diagnosis of RV infections, offering enhanced sensitivity and quantitative capabilities. While its application in clinical diagnostics is still emerging, dPCR holds significant potential for improving detection accuracy and understanding the epidemiology of RV infections [70,71].

### 3.3. Sequencing Methods

Sequencing technologies have become integral to the diagnosis and molecular characterization of RV infections, offering insights into viral evolution, reassortment, and transmission dynamics. Sanger sequencing—historically considered the gold standard in viral genomics—is widely used to genotype RV strains based on partial sequencing of the VP7 (G type) and VP4 (P type) genes. This method provides high accuracy and reliable results for single-strain infections, with read lengths typically ranging from 500 to 800 base pairs (bp). However, its limitations include low throughput, limited ability to resolve mixed infections, and restricted genomic coverage, making it less suitable for large-scale epidemiological studies or detection of reassortant strains [72,73].

Next-Generation Sequencing (NGS) technologies have significantly advanced RV diagnostics by enabling comprehensive, high-throughput analysis of the viral genome. The Illumina MiSeq platform is one of the most commonly used short-read NGS systems in RV research. It allows for deep sequencing of all 11 genomic segments directly from clinical fecal specimens, often following reverse transcription and cDNA synthesis. Studies have demonstrated that MiSeq can produce near-complete RV genomes with >90% coverage in samples with a sufficient viral load (cycle threshold ≤25), enabling the detection of mixed-genotype infections, point mutations, and reassortment events with high sensitivity. The multiplexing capability of MiSeq facilitates the simultaneous processing of multiple samples, making it ideal for outbreak investigations and longitudinal surveillance [73,74].

Third-generation sequencing platforms, such as Oxford Nanopore Technologies’ MinION, have emerged as powerful tools for rapid, real-time genomic analysis of RVs. Unlike Illumina sequencing, MinION provides long-read capabilities and does not require prior amplification or complex library preparation, making it particularly advantageous in field and low-resource settings. The nanopore sequencing approach yielded near-complete genome coverage and enabled accurate genotyping within a matter of hours [75]. Additionally, large-scale metagenomic studies using nanopore sequencing have demonstrated its capacity to recover full-length RV genomes from over 90% of positive samples, with segment-level identity comparable to that of Sanger sequencing [76]. Nanopore-based methods have been extended to veterinary surveillance, allowing for the identification and complete genomic characterization of RV groups A, B, C, and H in pigs, underscoring the versatility of this platform in One Health applications [77]. Although nanopore sequencing currently has a higher per-read error rate compared to Illumina, ongoing improvements in basecalling algorithms and error correction methods are rapidly closing this gap.

Table 5 presents the types of diagnostic test, the patient samples required, and the type of material detected for RV diagnosis [78,79,80]. Figure 4 provides a summary of the diagnostics tests used [78,79,80].

## 4. Symptomatic Treatment and Antiviral Therapy

Symptomatic treatment suffices in most cases. The medical approach may vary depending on the severity of dehydration; therefore, initial assessment of the patient’s general condition and the degree of dehydration is essential. The best indicators of dehydration include percentage of body weight loss, prolonged capillary refill time, abnormal skin turgor, and abnormal respiratory pattern [78].

### 4.1. Hydration and Dietary Management Based on ESPGHAN/ESPID Recommendations

For hydration, oral rehydration therapy (ORS) is the most preferred method for treating mild and moderate dehydration in children with AGE, both in home and hospital conditions [14]. Hypotonic fluids (250 mOsm/L or less) are utilized, reducing the number of vomiting episodes, diarrheal stools, and the need for intravenous rehydration therapy, without significantly increasing the risk of hyponatremia compared with standard ORS [14,15,16].

If oral rehydration is not feasible, enteral rehydration via the nasogastric (NG) route is the method of choice. Intravenous hydration is indicated in cases of shock, severe acidosis, worsening dehydration symptoms despite oral or enteral rehydration therapy, persistent vomiting, or intestinal obstruction [78]. For severe dehydration in children, an isotonic (0.9%) NaCl solution or Ringer’s lactate solution is used, whereas a dextrose-containing solution may be administered for maintenance therapy. In shock, intravenous administration of an isotonic crystalloid solution (0.9% saline or lactated Ringer’s solution) with a 20 mL/kg bolus is recommended. In the event of persistent hypotension after the administration of the first bolus, it is recommended to proceed with a second (or third, if necessary) bolus infusion of 20 mL/kg over a duration of 10 to 15 min, while concurrently investigating alternative etiologies of shock [78].

Both oral and intravenous hydration are comparably effective in children with moderate dehydration [79]. Additionally, compared with intravenous treatment, enteral hydration has been shown to reduce hospital stay duration and adverse effects [65]. It is important to transition to oral rehydration in children treated intravenously as soon as the indications for IV therapy subside [80].

Nutritional management in acute diarrhea involves continuing breastfeeding throughout rehydration, and for older children, an age-appropriate diet during or after initial rehydration (max 4–6 h).

Subsequent studies have highlighted how the composition of breast milk shapes susceptibility to rotavirus infection. Civra et al. investigated the antiviral potential of preterm human colostrum and its extracellular vesicles (EVs) against RVs. Colostrum samples from mothers of preterm infants (*N* = 10) showed significant antiviral activity in vitro, with EVs playing a major role in this effect by interfering with the early stages of viral replication [81]. In a large cross-country study analyzing more than 1600 proteins from 30 pathogens in nearly 900 milk samples, Campo et al. demonstrated that higher levels of RV-specific IgA in breast milk were associated with a delayed onset of infection, with the strongest signal directed against the outer capsid VP4 protein [82]. Superti et al. investigated the antiviral activity of several breast milk proteins—including α-lactalbumin, β-lactoglobulin, apo-lactoferrin (apo-Lf), and iron-saturated lactoferrin (Fe^3+^-Lf)—against RV using human erythrocytes and HT-29 cells. Of these, apo-Lf was the most effective, binding directly to viral particles and blocking their attachment to host cell receptors, thereby preventing both hemagglutination and viral entry [83]. Additional studies confirmed a dose-dependent inhibition of RV replication by both apo- and iron-saturated bovine lactoferrin (bLf), with sialic acid removal further amplifying this effect [84]. Beyond native lactoferrin, its derivative peptide, bovine lactoferricin (Lfcin-B), has also shown promise. In vivo experiments revealed that pre-treatment with Lfcin-B not only protected mice from RV-induced diarrhea but also significantly lowered intestinal viral loads. These findings suggest that Lfcin-B strengthens intestinal epithelial function and enhances nonspecific resistance to RV, positioning it as a candidate for pharmaceutical development or a functional food supplement to support mucosal immunity [85]. Future research should more extensively explore the protective mechanisms of breast milk components against RVs and their potential applications in clinical prevention and therapy.

High sucrose content and other sugars in fruit juices or beverages like cola can lead to osmotic diarrhea and are not recommended [86].

### 4.2. Pharmacological Therapy—ESPGHAN/ESPID Recommendations

Table 6 presents a summary of ESPGHAN/ESPID recommendations [37] from 2014 on the pharmacological management of acute diarrhea in children, alongside selected original research articles published after the guideline release (between 2014 and 2025), presented in Table 6 and Section 4.2.1, Section 4.2.2, Section 4.2.3 and Section 4.2.4. The selection was based on an advanced PubMed and Scopus search, focusing on studies related specifically to RV-associated AGE, with review articles excluded; only studies conducted in human populations were included, except for Section 4.2.5, where relevant in vitro and in vivo studies were also considered. The following keyword combinations were used in the search strategy: (rotavirus) AND (ondansetron); (rotavirus) AND (dexamethasone); (rotavirus) AND (dimenhydrinate); (rotavirus) AND (granisetron); (rotavirus) AND (metoclopramide); (rotavirus) AND (loperamide); (rotavirus) AND (diosmectite); (rotavirus) AND (kaolin); (rotavirus) AND (racecadotril); (rotavirus) AND (bismuth subsalicylate); (rotavirus) AND (zinc); (rotavirus) AND (probiotics); (rotavirus) AND (prebiotics); (rotavirus) AND (synbiotics); (rotavirus) AND (folic acid); (rotavirus) AND (gelatine tannate); (rotavirus) AND (oral immunoglobulin). Date of searching: 31 July 2025.

To date, the United States Food and Drug Administration (FDA) has not approved any antiviral drugs effective against RVs. Therefore, symptomatic treatment remains the main strategy in anti-RV therapy [90].

#### 4.2.1. Zinc

Table 7 presents a comprehensive overview of studies published between 2014 and 2025 that evaluate the clinical efficacy of zinc supplementation in the treatment of pediatric RV AGE [91,92,93].

Zinc alone or in combination with probiotics and selenium has been shown to improve clinical outcomes, reduce inflammation, and facilitate gut health in pediatric RV enteritis, as demonstrated by the studies referenced in Table 7.

#### 4.2.2. Probiotics Supplementation

Probiotic supplementation has shown promise in modulating the immune response during RV infections [94]. Table 8 summarizes recent clinical studies conducted between 2014 and 2025 that investigate the efficacy of probiotic therapy in the management of pediatric RV AGE. These studies encompass randomized controlled trials and interventional analyses evaluating various probiotic strains—both as monotherapies and in combination—administered to children with confirmed rotaviral infection [94,95,96,97,98,99,100,101,102,103].

As shown in Table 8, probiotics in children with RV diarrhea typically shorten the duration of diarrhea and hospital stay, improve immune function, and are safe with few side effects. Efficacy varies depending on probiotic strain, type of infection, and vaccination [94,95,96,97,98,99,100,101,102,103,104,105].

#### 4.2.3. Oral Immunoglobulins

Table 9 provides a concise summary of recent studies evaluating the clinical efficacy of immunoglobulin-based therapies in the management of pediatric RV AGE, encompassing both randomized controlled trials and observational case reports published between 2014 and 2025 [106,107,108].

Taken together, these findings support the potential of immunoglobulin therapy—particularly orally administered IgY antibodies—as a beneficial adjunct in reducing diarrhea duration, viral shedding, and improving mucosal immunity in otherwise healthy pediatric patients with RV infection [106,107,108].

#### 4.2.4. Natural Compounds—Quercetin

Quercetin, a plant-derived flavonoid, has demonstrated antiviral properties against RV, both in vitro and in vivo. It exerts its effects by inhibiting the NF-κB signaling pathway, which is activated during RV infection to promote viral replication and suppress apoptosis. By modulating this pathway, quercetin reduces viral protein expression and alleviates intestinal damage, leading to improved stool consistency and recovery of intestinal villi in animal models. Notably, quercetin exhibits minimal cytotoxicity, suggesting its potential as a safe adjunctive treatment in human infections [109].

#### 4.2.5. Antiviral Compounds

There are currently no approved antiviral drugs specifically targeting RV infections. Figure 5 illustrates the key stages of the RV life cycle alongside potential antiviral intervention points and corresponding compounds. Table 10 provides a comprehensive summary of these antiviral agents, including their classes, mechanisms of action, and the type of study in which they were evaluated [90,91,92,93,94,95,96,97,98,99,100,101,102,103,104,105,106,107,108,109,110,111,112,113,114,115,116].

Evidence indicates that RV can be inhibited at numerous stages of its life cycle, including viral entry, genome replication, viroplasm formation, and infectious particle genesis. Drugs that act early on like cordycepin, cyclosporine, and methyl-β-cyclodextrin (MβCD) interact with PI3K/Akt signaling, enhance type I interferon (IFN-I) responses, or disrupt cholesterol rafts, thereby inhibiting viral penetration [118,119,124,125]. During replication, compounds like 6-thioguanine (6-TG), brequinar and leflunomide, and metformin hydrochloride suppress viral RNA synthesis through metabolic or nucleotide pathway inhibition, and cellular microRNAs (miRNAs) like miR-525-3p and miRNA-7 contribute to antiviral immunity by targeting non-structural proteins (NSPs) [22,111,123,130,131].

Targeting viroplasm formation is particularly effective, with nitazoxanide, ursolic acid, and proteasome inhibitors blocking viral assembly by lipid droplet modulation and protein maturation inhibition [25,115,116,120,134,135]. During the following phases, compounds such as 18β-glycyrrhetinic acid and rapamycin exploit host signaling and autophagy to restrict particle formation [25,120].

However, most of the current evidence is derived from in vitro and animal studies, with only limited clinical data available. To establish efficacy and safety in humans, large-scale, randomized, double-blind, placebo-controlled trials remain essential.

## 5. Materials and Methods

This work was designed as a narrative review, drawing on a comprehensive analysis of research articles retrieved from major biomedical and multidisciplinary databases. Section 1, Section 2, Section 3 and Section 6 provide an in-depth analysis of studies available on PubMed, Google Scholar, Web of Science, Embase, and Scopus, identified using search terms related to: “rotavirus diarrhea”, “clinical diagnostic methods”, “innovative diagnostic methods”, “treatment of rotaviral diarrhea”, “prevention of viral infections”. While the primary focus was on the literature published within the past decade, the review spans a timeline from 1984 to 2025 to ensure a comprehensive, thorough perspective.

Section 4 presents a concise summary of the 2014 ESPGHAN/ESPID recommendations regarding the pharmacological management of acute diarrhea in children. This is followed by a critical appraisal of selected original research articles published between 2014 and 2025 (as shown in Table 6 and detailed in Section 4.2.1, Section 4.2.2, Section 4.2.3, Section 4.2.4 and Section 4.2.5). The selection of studies was based on a comprehensive and systematic literature search performed in PubMed and Scopus, targeting original research focused exclusively on RV-associated AGE in the pediatric population. Review articles were deliberately excluded to ensure analytical emphasis on primary clinical data. Date of searching: 31 July 2025.

Figure 1, Figure 2, Figure 3, Figure 4 and Figure 5 were created using GoodNotes 6 for iPad (v.6.3.55)—all based on the authors’ interpretation of synthesized data from multiple studies. Figure 4 was created using draw.io v27.0.5 (JGraph Ltd., Northampton, UK; draw.io AG, Zürich, Switzerland; operational headquarters: Wiesbaden, Germany).

## 6. Conclusions

RV is the leading cause of severe AGE in children under five, causing significant morbidity. While vaccination programs have reduced infection rates in high-income countries, RV remains a major health threat in low- and middle-income regions. Symptoms include diarrhea, vomiting, fever, and dehydration, with rare neurological complications.

Diagnosis primarily relies on stool testing using antigen detection methods like ELISA and ICA, and the more sensitive RT-PCR, which also enables viral genotyping. Although RT-PCR is the diagnostic gold standard, its cost and complexity limit its use in many settings. Antigen tests are favored for their short turnout time and affordability but can produce false positives, especially in recently vaccinated children. Accurate diagnosis is essential for clinical care, surveillance, and evaluating vaccine effectiveness. Recent advances in RV diagnosis include several highly sensitive and efficient methods. PAGE separates viral RNA segments and combined with silver staining, detects RV with high sensitivity and specificity. EIA offers rapid detection of RV antigens with improved sensitivity and specificity, providing results within minutes and performing well across different patient groups. dPCR is an emerging molecular technique that enables precise detection and quantification of low viral loads, useful for environmental surveillance and detection of viral mutations. Together, these innovations enhance RV detection accuracy and epidemiological monitoring.

We presented a comprehensive summary of current diagnostic and therapeutic approaches to RV AGE, with particular relevance to clinical practice. We outlined supportive and pharmacological treatments discussed in the recent literature and ESPGHAN guidelines, including hydration therapy, antiemetic drugs (ondansetron, dexamethasone, dimenhydrinate, granisetron, metoclopramide), antimotility and antiperistaltic agents (loperamide), adsorbents (diosmectite, diosmectite plus *Lactobacillus rhamnosus* GG, kaolin–pectin, attapulgite-activated charcoal), antisecretory agents (racecadotril, bismuth subsalicylate), zinc, probiotics, synbiotics, prebiotics, micronutrients (e.g., folic acid), gelatin tannate, and oral immunoglobulin therapy. Moreover, this review examines a broad spectrum of experimental antiviral agents with activity against distinct phases of the RV life cycle. These include natural bioactive compounds, immunomodulatory molecules, and targeted pharmacological inhibitors validated in preclinical settings. Among the most notable are Genipin, Cyclosporine, Cordycepin, Methyl-β-cyclodextrin, Genistein, Drebrin, ML241, Brequinar, Leflunomide, POL-P (*Portulaca oleracea* L.), Metformin hydrochloride, 6-Thioguanine (6-TG), MiR-525-3p, miRNA-7, Nitazoxanide, Ursolic acid, Molnupiravir, Deoxyshikonin, lipid metabolism regulators (e.g., TOFA, triacsin C, C75, A922500, betulinic acid, CI-976, PHB, isoproterenol/IBMX), proteasome inhibitors (MG132, bortezomib, and lactacystin), 18β-Glycyrrhetinic acid (18βGRA), RA839, and autophagy-inducing agents such as Rapamycin, LY294002, and BEZ235. Our work aims to support clinicians by providing an updated overview of diagnostic tools and therapeutic strategies, thereby contributing to improved patient care and evidence-based decision-making in the management of RV infections.

## Figures and Tables

**Figure 1 ijms-26-09175-f001:**
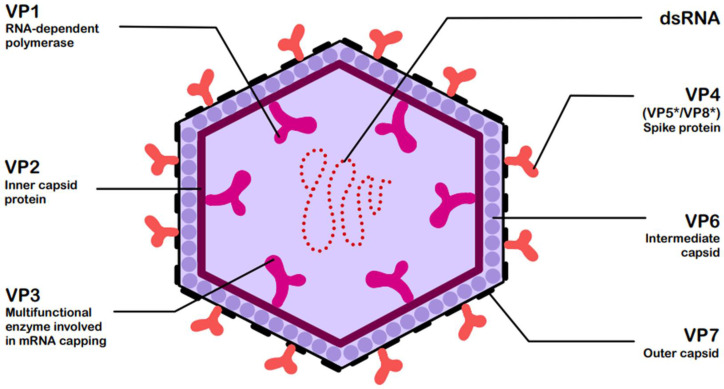
Structure of rotavirus (RV) with functions of structural viral proteins (VPs) [6,8]. The asterisk * denotes these trypsin-cleaved, activated fragments.

**Figure 2 ijms-26-09175-f002:**
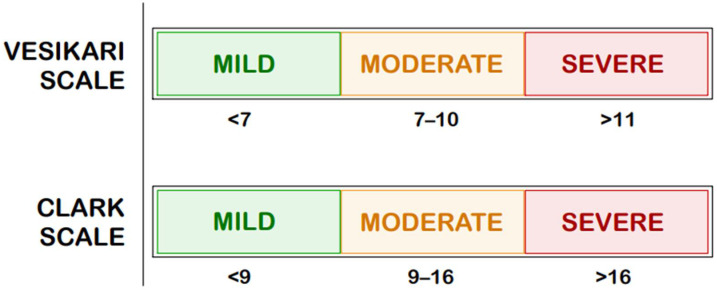
Comparison of Clark and Vesikari Scale Scores [17,18].

**Figure 3 ijms-26-09175-f003:**
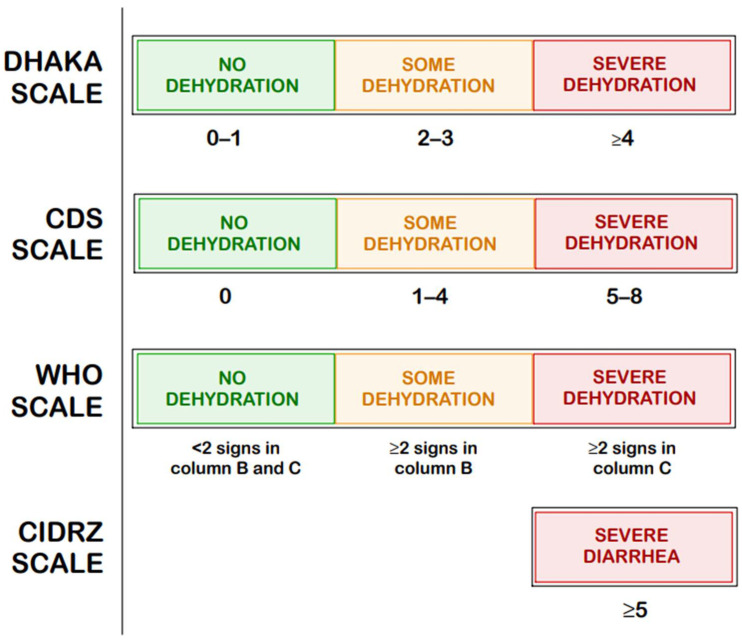
Comparison of the Clinical Dehydration Scale (CDS), The Dehydration: Assessing Kids Accurately (DHAKA) Scale, Centre for Infectious Disease Research in Zambia (CIDRZ) Scale, and the dehydration scale by the World Health Organization (WHO) [19,20,21,22,23].

**Figure 4 ijms-26-09175-f004:**
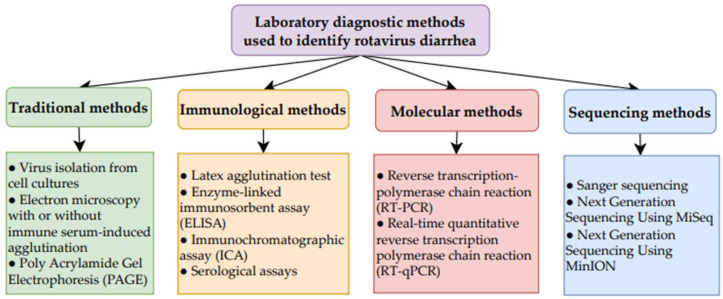
Summary of laboratory diagnostic methods used to identify RV diarrhea [78,79,80].

**Figure 5 ijms-26-09175-f005:**
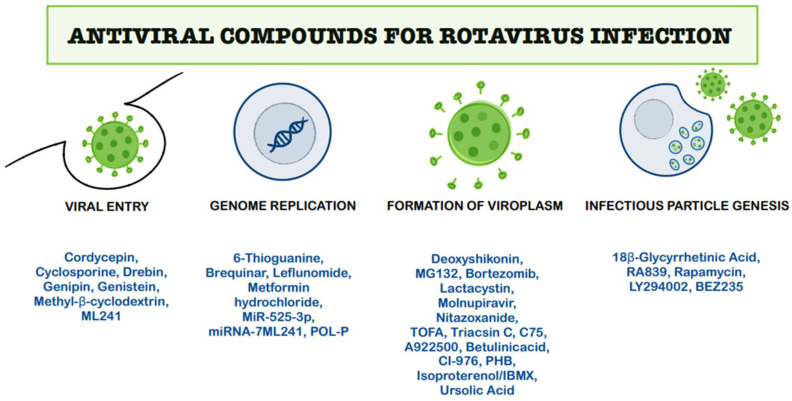
Schematic representation of the RV life cycle and potential antiviral targets [110,111,112,113,114,115,116,117,118,119,120,121,122,123,124,125,126,127,128,129,130,131,132,133,134,135,136].

**Table 1 ijms-26-09175-t001:** Comparison of parameters used in scales (the Vesikari Scoring System (VSS), the Clark scale, the Clinical Dehydration Scale (CDS), the Dehydration: Assessing Kids Accurately (DHAKA) scale, the Centre for Infectious Disease Research in Zambia (CIDRZ) scale, the 10- and 4-point Gorelick scale for dehydration, the World Health Organization (WHO) scale for dehydration) assessing the severity of rotavirus (RV) acute gastroenteritis (AGE) in Children [17,18,19,20,21,22,23].

The Vesikari Scoring System (VSS) [17]								
Characteristics/Points	1			2			3	
Number of stools/days	1–3			4–5			≥6	
Duration of diarrhea (days)	1–4			5			≥6	
Number of emesis events/day	1			2–4			≥5	
Duration of emesis (days)	1			2			≥3	
Rectal temperature (°C)	37.1–38.4			38.5–38.9			≥39	
Dehydration	-			1–5%			≥6%	
Treatment	Rehydration			Hospitalization			-	
The Clark Scale [18]								
Characteristics/Points	1			2			3	
Number of stools/days	2–4			5–7			≥8	
Duration of diarrhea (days)	1–4			5–7			≥8	
Number of emesis events/day	1–3			4–6			≥7	
Duration of emesis (days)	2			3–5			≥6	
Rectal temperature (°C)	38.1–38.2			38.3–38.7			≥38.8	
Temperature duration (days)	1–2			3–4			≥5	
Behavioral symptoms	Irritable/less playful			Lethargic/listless			Seizures	
Duration of behavioral symptoms (days)	1–2			3–4			≥5	
Clinical Dehydration Scale (CDS) [19]								
Characteristics/Points	1			2			3	
General appearance	Normal			Thirsty, restless, or lethargic but irritable when touched			Cold, drowsy, limp, sweaty, or comatose	
Eyes	Normal			Slightly sunken			Extremely sunken	
Mucous membrane (tondue)				Sticky			-	
Tears	Normal			Decreased tears			Absent tears	
The Dehydration: Assessing Kids Accurately (DHAKA) Scale [20]								
Characteristics/Points	0			2			4	
General appearance	Normal			Restless, irritable			Lethargic, unconscious	
Tears	Normal			Decreased			Absent	
Skin pinch	Normal			Slow			Very slow	
Respirations	Normal			Deep			-	
Centre for Infectious Disease Research in Zambia (CIDRZ) Scale [21]								
Characteristics/Points	1		2		3			Characteristics/Points
Frequency of vomiting			2–3 episodes/day		4–5 episodes/day			≥6 episodes/day
General appearance	-		-		Restless/irritable			Lethargic
Skin pinch test	Normal (instant recoil)				Slow recoil (>2 s)			Very slow
Tears	Present		-		Absent			-
Respirations	Normal		-		Deep			-
The 10- and 4-point Gorelick Scale for dehydration (4-point scale with) [22]								
Characteristics		No or minimal dehydration				Moderate to severe dehydration		
General appearance		Alert				Restless, lethargic, unconscious		
Tears		Present				Absent		
Mucous membranes		Moist				Dry, very dry		
Capillary refill		Normal				Prolonged or minimal		
Eyes		Normal				Sunken		
Quality of pulses		Normal				Thready; weak or impalpable		
Heart rate		Normal				Tachycardia		
Urine output		Normal				Reduced; not passed in many hours		
Breathing		Present				Deep; deep and rapid		
Skin elasticity		Instant recoil				Recoil slowly; recoil > 2 s		
The World Health Organization (WHO) scale for dehydration [23]								
Characteristics/Points	A			B			C	
General appearance	Good, active			Irritable, restless			Lethargic or unconscious	
Eyes	Normal			Sunken			Sunken	
Thirst	Not thirsty, drinks normally			Thirsty, drinks eagerly			Not able to drink or drinks poorly	
Skin turgor	Instant recoil			Slow return to normal			Very slow return to normal	

**Table 2 ijms-26-09175-t002:** Comparison of parameters used in the Vesikari Score System (VSS) and the Modified Vesikari Severity Score (MVSS) [30,33].

Symptom or Sign	Vesikari Score System (VSS)	Modified Vesikari Severity Score (MVSS)
Max. no. diarrheal stools/24 h		
1–3	1	1
4–5	2	2
≥6	3	
Duration of diarrhea in days		
1–4	1	1
5	2	2
≥6	3	
Max. no. vomiting episodes/24 h		
1	1	1
2–4	2	
≥5	3	2
Duration of vomiting in days		
1	1	1
2	2	
≥3	3	2
Fever		
<37.0 °C	0	0
37.1–38.4 °C	1	1
38.5–38.9 °C	2	2
≥39 °C	3	3
Dehydration		
Little to mild	1	1
Mild to moderate (1–5%)	2	2
Severe (≥6%)	3	3
Treatment		
None	0	0
Rehydration	1	1
Hospitalization	2	2
Duration of fever in days		
0	Not assessed	0
1–3	Not assessed	1
≥3	Not assessed	2
Gastrointestinal hemorrhage		
None	Not assessed	0
Occult blood in stool only	Not assessed	1
Gross bloody stool	Not assessed	2
Convulsion		
None	Not assessed	0
Yes	Not assessed	1
With recurrence	Not assessed	2
Assessment of abdomen pain or flatulence		
None	Not assessed	0
Flatulence	Not assessed	1
Irritability or pain	Not assessed	2
Total Score	Max 20	Max 24

In the Vesikari score, a total score of <7 indicates mild disease, 7–10 indicates moderate disease, and >10 indicates severe disease. In the modified Vesikari score, <8 is classified as mild, 8–11 as moderate, and >11 as severe.

**Table 3 ijms-26-09175-t003:** Comparative performance of dehydration scoring tools in predicting clinical outcomes in children under five years of age [21].

Scoring Scale	Area Under Curve (AUC)	95% CI
Vesikari Scale	0.26	−0.17, 0.56
Clark Scale	0.18	−0.17, 0.56
DHAKA Score	0.59	0.11, 1.00
CIDRZ Scale	0.59	−0.13, 0.39

Abbreviations: AUC—Area Under Curve; DHAKA—Dehydration: Assessing Kids Accurately; CIDRZ—Centre for Infectious Disease Research in Zambia; CI—Confidence Interval.

**Table 4 ijms-26-09175-t004:** Diagnostic performance of clinical dehydration scales in children aged 1 month to 5 years with diarrhea [38].

**Dehydration Scale**	**AUC**	**95% CI**
WHO Scale (≥5% dehydration)	0.71	0.65–0.77
Gorelick 4-point	0.71	0.63–0.78
Gorelick 10-point	0.74	0.68–0.81
CDS	0.54	0.45–0.63

Abbreviations: WHO—World Health Organization; CDS—Clinical Dehydration Scale; AUC—Area Under Curve; CI—Confidence Interval.

**Table 5 ijms-26-09175-t005:** Types of diagnostic tests, including the type of specimen collected from the patient and the name of the detected RV particle [78,79,80].

Type of Test	Suitable Specimen	Target
Culture	Fluid stool	Bottles after trypsin pre-treatment. RV has been cultured in primary African green monkey kidney and MA104 cells.
Latex Agglutination	Fluid stool	Detection of the VP6 antigen of group A RVs.
Enzyme immunoassay (EIA)	Fluid stool	Detection of the VP6 antigen of group A RVs.
Immunochromatographic Tests (ICAs)	Fluid stool	A strip with immobilized monoclonal antibodies against the VP6 protein of RV.
Reverse Transcriptase Polymerase Chain Reaction (RT-PCR)	Fluid stool	PCR assays for RV detection commonly target the VP6 gene, while genotyping assays focus on the VP7 and VP4 genes.
Serological Tests	Clotted blood 1 week after illness	RV-specific IgM appears in serum about one week after symptom onset. Serum IgA is the primary serologic marker for infection and the most reliable indicator of reinfection in vaccine studies.
Sanger Sequencing	Fluid stool (RNA extracted)	Partial sequencing of VP4 and VP7 genes for genotyping; highly accurate for identifying known strains but low throughput.
Next-Generation Sequencing (NGS)—Illumina MiSeq	Fluid stool (RNA extracted, cDNA synthesized)	Whole genome sequencing of all 11 segments; detects mixed infections, point mutations, and reassortment events. Requires relatively high viral load for optimal coverage.
Next-Generation Sequencing (NGS)—Oxford Nanopore MinION	Fluid stool (RNA extracted)	Long-read sequencing of all RV segments; useful for rapid, real-time detection and full genome assembly directly from clinical specimens. Effective in identifying rare or reassortant strains even in low-resource settings.

Abbreviations: RV—rotavirus.

**Table 6 ijms-26-09175-t006:** ESPGHAN/ESPID recommendations (2014) regarding pharmacological treatment of AGE in children and original studies published thereafter related to RV-associated AGE (excluding review articles) [37].

Name of Drug	ESPGHAN/ESPID (2014) Recommendation	ESPGHAN/ESPID (2014) Strength of Recommendation/Quality of Evidence	Research Associated with RV (If Possible) Infection Published from 2014 to 2025
Ondansetron (antiemetic drug)	Administered either orally or intravenously—appears potentially effective in managing vomiting in young children with AGE; nevertheless, confirmation of its safety in pediatric populations is required prior to issuing conclusive recommendations.	(II, B) * (strong recommendation, low-quality evidence) **	In a randomized trial involving 104 children with AGE, a single oral dose of ondansetron (0.15 mg/kg) significantly reduced symptom duration in RV-positive cases (median: 2 days, *p* = 0.014) and decreased diarrheal episodes in children symptomatic for >3 days (*p* = 0.028). These findings suggest that ondansetron, by attenuating gastrointestinal symptoms, may facilitate oral rehydration and potentially reduce the need for hospitalization in children with RV AGE [87].
Dexamethasone, dimenhydrinate, granisetron, and metoclopramide (antiemetics drugs)	The use of other antiemetic agents is not supported by current evidence.	(II, B) * (strong recommendation, low-quality evidence) **	No new research available on PubMed and Scopus from 2014 to 2025.
Loperamide (antimotility and antiperistaltic drug)	Not recommended in the management of AGE in children.	(II, B) * (strong recommendation, very low-quality evidence) **	No new research available on PubMed and Scopus from 2014 to 2025.
Diosmectite	Can be considered.	(II, B) * (weak recommendation, moderate-quality evidence) **	No new research available on PubMed and Scopus from 2014 to 2025.
Diosmectite Plus LGG	Since Smectite plus LGG and LGG alone provide similar therapeutic outcomes in young children with AGE, using both together is not supported.	(II, B) * (weak recommendation, low-quality evidence) **	No new research available on PubMed and Scopus from 2014 to 2025.
Other absorbents (kaolin–pectin and attapulgite-activated charcoal)	Currently not recommended for managing AGE in pediatric patients.	(III, C) * (weak recommendation, very low-quality evidence) **	No new research available on PubMed and Scopus from 2014 to 2025.
Racecadotril (antisecretory drugs)	Currently not recommended for managing AGE in pediatric patients.	(II, B) * (weak recommendation, moderate-quality evidence) **	In two randomized, double-blind, placebo-controlled trials conducted in community and hospital settings in Vellore, India, racecadotril (1.5 mg/kg TID for 3 days) was evaluated in children aged 3–59 months with acute diarrhea. A total of 326 children completed the trials. Racecadotril showed no significant benefit over placebo in reducing diarrheal duration, stool output, or fluid intake, regardless of RV status. Median diarrhea duration remained comparable between groups in both settings. These findings do not support the use of racecadotril for managing AGE in young children [88].
Bismuth Subsalicylate	Currently not recommended for managing AGE in pediatric patients.	(III, C) * (strong recommendation, low-quality evidence) **	No new research available on PubMed and Scopus from 2014 to 2025.
Zinc	The therapeutic value of zinc in treating AGE is mainly observed in children from developing countries; in regions with sufficient zinc intake, its benefit is minimal.	(I, A) * (strong recommendation, moderate-quality evidence) **	Described in Section 4.2.1.
Probiotics	Probiotics, when used with oral rehydration, help reduce the severity of symptoms in children with AGE (I, A). LGG and *S. boulardii* are recommended as adjuncts to rehydration (I, A)	(I, A) * (strong recommendation, moderate-quality evidence) about probiotics ** and (I, A) * (strong recommendation, low-quality evidence) about LGG and *S. boulardii* **	Described in Section 4.2.2.
Synbiotics	Not recommended until supported by well-documented studies.	(II, B) * (weak recommendation, low-quality evidence) **	In a study involving 69 children aged 6–59 months with acute diarrhea, 34 received a synbiotic formulation comprising *L. casei* (4 × 10^8^ CFU), *L. rhamnosus* (3.5 × 10^8^), *S. thermophilus* (1 × 10^8^), *B. breve* (5 × 10^7^), *L. acidophilus* (5 × 10^7^), *B. infantis* (4 × 10^7^), *L. bulgaricus* (1 × 10^7^), and 990 mg of FOS (1 × 10^9^ CFU total) once daily for five days, alongside ORS/IV rehydration and zinc therapy, while 35 children served as controls (Placebo + ORS/IV rehydration therapy + Zn). Significant decrease in diarrhea recovery time. No impact on hospital stay [89].
Prebiotics	Currently not recommended for managing AGE in pediatric patients.	(II, B) * (weak recommendation, low-quality evidence) **	No new research available on PubMed and Scopus from 2014 to 2025.
Micronutrients like folic acid	Currently not recommended for managing AGE in pediatric patients.	(II, B) * (weak recommendation, very low-quality evidence) **	No new research available on PubMed and Scopus from 2014 to 2025.
Gelatine Tannate	Currently not recommended for managing AGE in pediatric patients.	(III, C) * (weak recommendation, very low-quality evidence) **	No new research available on PubMed and Scopus from 2014 to 2025.
ORV	Can be considered for hospitalized children with RV AGE.	(III, C) * (weak recommendation, very low-quality evidence) **	Described in Section 4.2.3.

* Strength of evidence and grade of recommendations supporting the ESPGHAN/ESPID guidelines for the management of acute gastroenteritis in children in Europe, as formulated in 2008 and updated in the 2014 recommendations. Strength of evidence: I—strong evidence from ≥1 systematic review(s) of well-designed RCTs; II—strong evidence from ≥1 properly designed RCT(s) of appropriate size; III—evidence from well-designed trials without randomization, single group pre–post, cohort, time series, or matched case–control studies; IV—evidence from well-designed trials, nonexperimental studies from >1 center or research group. GRADE: A—supported by level I evidence, highly recommended; B—supported by level II evidence, recommended; C—supported by level III evidence, recommended; D—supported by level IV and level V evidence; the consensus route would have to be adopted. ** GRADE system, as applied in the 2014 ESPGHAN/ESPID guidelines. Quality of evidence: high quality—further research is unlikely to change our confidence in the estimate of effect; moderate quality—further research is likely to have an important impact on our confidence in the estimate of effect and may change the estimate; low quality—further research is extremely likely to have an important impact on our confidence in the estimate of effect and is likely to change the estimate; very low quality—any estimate of effect is extremely uncertain. Grade of recommendation: strong—when the desirable effects of an intervention clearly outweigh the undesirable effects, or they clearly do not; weak—when the tradeoffs are less certain (either because of the low quality of evidence or because the evidence suggests that desirable and undesirable effects are closely balanced). Abbreviations: RV—rotavirus; ESPGHAN/ESPID—European Society for Pediatric Gastroenterology, Hepatology, and Nutrition/European Society for Pediatric Infectious Diseases Evidence-Based Guidelines; AGE—acute gastroenteritis; SIgA—serum immunoglobuline A; LGG—*Lactobacillus rhamnosus* GG; *L. casei*—*Lactobacillus casei*; *S. thermophilus*—[..]; *B. breve*—*Bifidobacterium breve*; *L. acidophilus*—*Lactobacillus acidophilus*; *B. infantis*—*Bifidobacterium infantis*; *L. bulgaricus*—*Lactobacillus bulgaricus*; CFU—colony-forming unit; FOS—fructooligosaccharides; ORV/IV—oral/intra venous; GRADE—Grading of Recommendations, Assessment, Development, and Evaluations.

**Table 7 ijms-26-09175-t007:** Overview of research assessing the efficacy of zinc therapy in pediatric RV AGE published between 2014 and 2025 [91,92,93].

Study Short Methodology	Type of Study	Key Findings	Authors and the Date
In a randomized study of 103 pediatric patients with RV enteritis, participants received either standard therapy alone (*n* = 52) or standard therapy plus zinc gluconate supplementation for 10 days (*n* = 51). Clinical outcomes were assessed at 72 h, including symptom resolution and recovery of extra-intestinal organ involvement. A 3-month follow-up evaluated diarrhea recurrence.	Randomized Controlled Trial	Zinc supplementation significantly improved clinical outcomes in infants with RV enteritis, with a higher response rate (90% vs. 75%, *p* < 0.05) and shorter duration of diarrhea, fever, and vomiting compared to standard therapy alone (*p* < 0.05). Moreover, zinc reduced both the recurrence and severity of post-treatment diarrhea over a 3-month follow-up (*p* < 0.05).	Jiang et al. (2016) [91]
A total of 85 patients with RV AGE were divided into two groups based on treatment strategy: the control group (*n* = 42) received probiotics alone, while the experimental group (*n* = 43) received a combination of probiotics with zinc and selenium. Clinical efficacy, stool frequency, and adverse event incidence were evaluated to determine therapeutic effectiveness.	Original Article	Combined probiotic, zinc, and selenium therapy significantly enhanced clinical efficacy compared to probiotics alone (88.4% vs. 50%, *p* < 0.05), with faster symptom resolution and greater reductions in stool frequency. Post-treatment levels of myocardial enzymes (CK, CK-MB, AST) and inflammatory markers (IL-6, IL-8, hsCRP) were markedly lower in the combination group (*p* < 0.05).	Cai et al. (2022) [92]
This study included 45 pediatric patients hospitalized with RV enteritis. Patients were assigned to three groups (*n* = 15 each): untreated controls, conventional therapy (oral smectite), and conventional therapy plus oral zinc gluconate. Fecal samples were collected 6 h post-fasting. Treatment lasted 7 days, with age-adjusted dosing for both smectite and zinc.	Original Research Article	Zinc combined with conventional therapy improved clinical recovery in pediatric RV enteritis, shortening symptom duration and enhancing gut microbiota diversity. Beneficial genera (e.g., *Faecalibacterium*, *Bacteroides*) increased, while inflammatory markers (IL-6, TNF-α, CRP) showed negative correlations with key commensals. Zinc adjunctive therapy accelerates recovery, supports microbiota balance, and may reduce inflammation, informing clinical management strategies.	Xu et al. (2023) [93]

Abbreviations: RV—rotavirus; CK—creatine kinase; CK-MB—creatine kinase–MB isoenzyme; AST—aspartate aminotransferase; IL-6—interleukin 6; IL-8—interleukin 8; hsCRP—high-sensitivity C-reactive protein; TNF-α—tumor necrosis factor alpha; AGE—acute gastroenteritis.

**Table 8 ijms-26-09175-t008:** Overview of research assessing the efficacy of probiotics therapy in pediatric RV AGE published between 2014 and 2025 [94,95,96,97,98,99,100,101,102,103].

Study Short Methodology	Type of Probiotics	Population	Primary Outcomes	Key Findings	Authors and the Date
Single-center, open-label, randomized, controlled trial included 159 patients (age range, 3 mo to 14 yo).	*Bacillus mesentericus* TO-A (1.1 × 10^7^ CFU), *Clostridium butyricum* TO-A (2.0 × 10^7^ CFU), *Enterococcus faecalis* T-110 (3.17 × 10^8^ CFU) (BIO-THREE) 3 times daily for 7 days	61 children (RV and Salmonella)	Duration of diarrhea, duration of fever, length of hospital stay	Probiotic therapy significantly shortened the duration of diarrhea and fever and reduced hospitalization time compared to control. Benefits were observed in both RV and salmonella infections, indicating broad therapeutic potential (*p* < 0.0001).	Huang et al. (2014) [94]
An open-label, randomized, controlled trial included 200 patients with AGE, aged between 6 months and 5 years.	LGG in dose of 10 × 10^9^ CFU/day for five days (LGG group) or no probiotic medication (control group)	100 children in control group and 100 children in LGG group	Duration of diarrhea, number of stools per day	LGG administration in children with AGE significantly shortened duration of diarrhea compared to the control group [60 (54–72) h vs. 78 (72–90) h; *p* < 0.001]. The number of stools per day was significantly lower in the LGG group compared with the control group (*p* < 0.001).	Aggarwal S et al. (2014) [95]
Children with AGE aged 6 months to 5 years, testing positive for either RV in stool (coinfections were excluded), were randomized to LGG (ATCC 53103) or placebo, once daily for 4 weeks. Baseline demographic and clinical details were obtained.	LGG (ATCC 53103) 1 × 10^10^ CFU	140 children with acute AGE (RV positive subgroup)	Recurrence of diarrhea, intestinal function/permeability, serum IgG levels	In children with RV AGE, LGG reduced recurrent episodes (25% vs. 46%; *p* = 0.048) and impaired intestinal function (48% vs. 72%; *p* = 0.027). A significant rise in anti-RV IgG was observed (2215 vs. 456 EU; *p* = 0.003).	Sindhu et al. (2014) [96]
In vitro antiviral activities of probiotic isolates on RV infection were investigated in the Vero cell using a plaque reduction assay. Then several probiotic strains with the high antiviral activity were chosen for further clinical trials. Twenty-nine pediatric patients who presented with symptoms of viral AGE were enrolled in a double-blind trial and randomly assigned at admission to receive six probiotic strains.	*Bifidobacterium longum*, *B. lactis*, *Lactobacillus acidophilus*, *L. rhamnosus*, *L. plantarum*, and *Pediococcus pentosaceus* at a dose of 10^9^ CFU/g or a comparable placebo twice daily for 1 week.	Children with RV AGE; cell line studies	Duration of diarrhea (clinical); inhibition of viral replication (in vitro)	In vitro results confirmed inhibition of RV replication by the probiotics. Clinically, children treated with probiotics experienced a statistically significant reduction in the duration of diarrhea.	Lee et al. (2015) [97]
Single-blind, placebo-controlled, randomized trial.	Combined product: *L. casei*, *L. rhamnosus*, *S. termophilus*, *B. breve*, *L. acidophilus*, *L. bulgaricus*, *B. infantis* + ORS/IV rehydration therapy ORS/IV rehydration therapy Dosage: 1 × 10^9^ CFU/g 2×/day for 5 days	64 children with confirmed RV AGE	Duration of diarrhea, adverse effects	LGG group had significantly shorter mean duration of diarrhea (4.1 vs. 6.2 days). Findings confirm symptomatic benefit of LGG in acute RV.	Sobouti et al., 2016 [98]
Children (3 months to 5 years) with WHO-defined acute watery diarrhea and RV positive stool (*n* = 60) were randomized into intervention (*n* = 30) and control (*n* = 30) groups. The intervention group received SB.	SB (500 mg/day) for 5 days	100 children with confirmed RV	Diarrhea duration, safety	*S. boulardii* significantly reduced diarrhea duration. Safe and effective adjunct to standard therapy in acute RV.	Das et al. (2016) [99]
The children were randomly divided into the study’s two treatment groups: three days of the oral administration of a probiotics formula containing both *Bifidobacterium longum* BORI and *Lactobacillus acidophilus* AD031 or a placebo and the standard therapy for diarrhea.	*Bifidobacterium longum* BORI (2 × 10^10^ CFU/g) and *Lactobacillus acidophilus* AD031 (2 × 10^9^ CFU/g)	Infants with RV infection	Diarrhea duration, viral load	When the 57 cases completed the protocol, the duration of the patients’ diarrhea was significantly shorter in the probiotics group (4.38 ± 1.29, *N* = 28) than the placebo group (5.61 ± 1.23, *N* = 29), with a *p*-value of 0.001. There were no serious, adverse events and no differences in the frequency of adverse events in both groups. Probiotic combination reduced both diarrhea duration and viral shedding. Clinically and virologically beneficial.	Park et al. (2017) [100]
Double-blind, placebo-controlled randomized controlled trial.	*L. acidophilus* (2 daily oral doses of 2 × 10^8^ for 5 days) *+* ORS + Zn + antimicrobials were needed Placebo + ORS + Zn + antimicrobials were needed.	300 Vietnamese children with acute watery diarrhea (RV-positive subgroup)	Duration of diarrhea, treatment failure	No significant difference in duration of diarrhea between treatment and placebo groups. Subgroup analysis (RV cases) also showed no benefit.	Hong Chau et al., 2018 [101]
Randomized, controlled clinical trial with retrospective comparison, conducted in hospitalized children with RV AGE.	*Lactobacillus plantarum* LRCC5310	50 children (15 probiotic group, 8 control, 27 retrospective control) hospitalized with confirmed RV AGE	Duration of diarrhea, number of stools, viral load in stool samples	LRCC5310 significantly reduced diarrhea frequency and duration (*p* = 0.033, *p* = 0.003, and *p* = 0.012, respectively), improved Vesikari scores (*p* = 0.076, *p* = 0.061, and *p* = 0.036, respectively), and inhibited viral replication compared with controls; no adverse effects reported.	Shin et al. (2020) [102]
Randomized controlled trial comparing racecadotril (1.5 mg/kg, given three times daily for 7 days) alone vs. racecadotril plus *Lactobacillus reuteri* in children with rotavirus enteritis.	*Lactobacillus reuteri* (administered daily at 1 × 10^8^ CFU for 7 days)	85 children with confirmed RV AGE (43 control, 42 probiotic + racecadotril)	RV conversion rate, intestinal mucosal barrier function, immune response (CD4+, CD8+), intestinal microbiota composition	Combination therapy led to significantly higher RV clearance rates at days 3, 5, and 7 (*p* < 0.05), reaching 61.90%, 76.19%, and 92.86%, respectively, improved mucosal barrier function, increased CD4+ and beneficial microbiota, and reduced endotoxins, AGEs, D-lactic acid, and CD8+ levels compared with controls.	He et al. (2025) [103]

Abbreviations: RV—rotavirus; Zn—zinc; LRCC5310—*Lactobacillus plantarum* strain; *p*—*p*-value (statistical significance); WHO—World Health Organization; AGE—acute gastroenteritis.

**Table 9 ijms-26-09175-t009:** Overview of research assessing the efficacy of Oral immunoglobulins therapy in pediatric RV AGE published between 2014 and 2025 [106,107,108].

Study Short Methodology	Type of Oral Immunoglobuline with Dosage	Type of Study	Key Findings	Authors and the Date
A total of 100 pediatric patients with RV enteritis divided into control and immunoglobulin treated group. All patients received fluid replacement.	A randomized trial included two groups of children aged 3 months to 3 years. The control group (*n* = 50) received oral placebo, while the treatment group (*n* = 50) received anti-RV IgY from egg yolk (1 g, three times daily for 3 days). Baseline characteristics, including age, sex, and pre-treatment diarrhea frequency, showed no significant differences between groups (*p* > 0.05).	Clinical trial, randomized controlled trial	Orally administered anti-RV IgY significantly reduced diarrhea duration (4.5 ± 0.92 vs. 5.8 ± 1.68 days, *p* = 0.015), decreased stool frequency (*p* < 0.05), increased fecal SIgA levels with earlier doubling (day 3 vs. day 5), and lowered viral shedding compared to placebo (*p* < 0.05).	Xie et al. (2015) [106]
A total of 4 pediatric hematopoietic stem cell transplant patients with confirmed RV infection.	Patients received human immunoglobulin (Gamunex-C, 10%, Grifols Therapeutics, LLC, Research Triangle Park, NC, USA). Two with acute myeloid leukemia (ages 10 months and 5 years) and one with Shwachman–Diamond syndrome (14 years) received 20 mg/kg/dose QID. A patient with Wiskott–Aldrich syndrome (14 months) received 76 mg/kg/dose QID.	Case report	In three of four treatment episodes, stool frequency and/or consistency improved within a median of three days after starting enteral immunoglobulin. Symptom resolution was faster than in historical controls. One patient was later diagnosed with GI graft-versus-host disease after 22 months. These results indicate potential benefits of oral immunoglobulin for managing RV diarrhea in HSCT recipients.	Williams et al. (2015) [107]
A total of 36 pediatric allogeneic HCT recipients with total 49 discrete episodes of RV infection (positive stool RV antigen assay).	Nitazoxanide and/or enterally administered immunoglobulins. One patient received 16 doses of immunoglobulin over 5 days at 0.34 g/kg/dose; all other patients received a single initial dose.	Retrospective single-center study	Median duration of diarrhea: 17.5 days (range 4–122). After initiation of treatment, the median duration of clinical symptoms are 11 days (nitazoxanide), 23 days (immunoglobulins), and 26 days (combination). No adverse effects observed. Efficacy not confirmed.	Flerlage et al., (2018) [108]

Abbreviations: RV—rotavirus.

**Table 10 ijms-26-09175-t010:** Antiviral compounds evaluated for RV infection in this review [110,111,112,113,114,115,116,117,118,119,120,121,122,123,124,125,126,127,128,129,130,131,132,133,134,135,136].

Stage of Viral Life Cycle	Compound	Class of Inhibitor	Mechanism of Action	Type of Study
Viral entry	Cordycepin	Adenosine analog	Modulates PI3K/Akt pathway, promotes apoptosis, and suppresses viral replication.	Investigational in vitro and in vivo study (BALB/c infant mice) [119].
	Cyclosporine	Cyclic peptide (host factor modulator)	Increases expression of type I interferons, enhancing antiviral response.	Investigational in vitro and in vivo study (BALB/c infant mice) [118].
	Drebin	Cytoskeleton-associated protein	Binds VP4 and its VP5 fragment, hindering viral entry into host cells.	In vitro study; also evaluated in vivo: human primary enteroids and C57BL/6 mouse model [128].
	Genipin	Aglycone (from geniposide)	Inhibits viral attachment and penetration (early stage) and also assembly and release (late stage).	In vitro studies [113].
	Genistein	Flavonoid, tyrosine kinase inhibitor	Inhibits integrin phosphorylation, reduces RV binding affinity to integrins and entry.	In vitro studies [127].
	Methyl-β-cyclodextrin	Pharmacological cholesterol-sequestering agent	Disrupts cholesterol rafts and blocks RV receptor-mediated endocytosis.	In vitro studies [124,125].
Viral entry, Genome replication	ML241	Small-molecule MAPK pathway inhibitor	Inhibits ERK1/2 phosphorylation and IκBα/NF-κB signaling, suppressing viral entry, replication, and cytotoxicity.	In vitro and in vivo study using BALB/c infant mice [121].
Genome replication	6-Thioguanine (6-TG)	Thio-analog of guanine	GTP-Rac1 inhibitor	In vitro study; tested in vivo: human enteroids; 6-TG has been clinically used since the 1950s as an anticancer and immunosuppressive agent [22].
	Brequinar	Quinolinecarboxylic acid	Inhibits pyrimidine synthesis via DHODH inhibition.	In vitro studies [111].
	Leflunomide	Isoxazole derivative	Inhibits pyrimidine synthesis via DHODH inhibition.	In vitro studies [111].
	Metformin hydrochloride	Biguanide (metabolic regulator)	Reduces viral mRNA levels and RV replication in Caco-2 cells and intestinal organoids.	In vitro and in vivo study using BALB/c infant mice [123].
	MiR-525-3p	Cellular microRNA	Binds the 3′ UTR of RV NSP1, enhancing interferon and cytokine levels.	In vitro studies [130].
	miRNA-7	Cellular microRNA	Targets RV gene segment 11 encoding NSP5, disrupting viroplasm formation.	In vitro and in vivo study using BALB/c infant mice [131].
	POL-P (*Portulaca oleracea* L.)	Polysaccharide (plant-derived immunomodulator)	Upregulates IFN-α expression, suppressing viral replication.	In vitro studies [112].
Formation of Viroplasm	Deoxyshikonin	Naphthoquinone derivative	Activates SIRT1, ac-FoxO1, and Rab7; lowers VP6 expression and viral titers; induces autophagy and oxidative stress.	In vitro studies [122].
	MG132, bortezomib, and lactacystin	Tripeptide aldehyde, dipeptidylboronate, and antibiotic, respectively	Proteasome inhibitors; disrupt VP formation and alter VP1 localization.	In vitro studies [134,135].
	Molnupiravir	Cytidine nucleoside analog	Inhibits viroplasm formation without altering structure.	In vitro studies [117].
	Nitazoxanide	Thiazolide	Inhibits VP7 maturation, disrupts viroplasm assembly, and impairs viral morphogenesis.	Clinical trial for RV treatment; clinically approved anti-infective treatment [115].
	TOFA, triacsin C, C75, A922500, betulinic acid, CI-976, PHB, and isoproterenol/IBMX	Pharmacological enzyme inhibitors of the lipid metabolism pathway	Modulates lipid droplet biogenesis and degradation.	In vitro studies [133].
	Ursolic acid	Triterpenoid	Reduces lipid droplet availability, limiting VP formation.	In vitro studies [116].
Infectious particle genesis	18β-Glycyrrhetinic acid (18βGRA)	Aglycone (PI3K/Akt pathway; antiviral activity)	Inhibits viral replication by modulating apoptosis and signaling pathways.	In vitro studies, tested in a mouse model (C57BL/6 male mice) [120].
	RA839	Small molecule	Activates the Nrf2/ARE pathway, enhancing the cellular redox response.	In vitro studies [126].
	Rapamycin, LY294002, and BEZ235	Macrolide, morpholine-containing chemical compound, and imidazoquinoline derivative, respectively	mTOR and PI3K inhibitors; activate the autophagy cascade.	In vitro study; tested in vivo: human enteroids; rapamycin is FDA-approved for transplant recipients [25].

Abbreviations: RV—rotavirus; VP—viral protein; VP1/VP4/VP5*/VP6/VP7—structural viral proteins of rotavirus; IFN—interferon; NF-κB—nuclear factor kappa-light-chain-enhancer of activated B cells; ERK1/2—extracellular signal-regulated kinases 1 and 2; IκBα—inhibitor of kappa B alpha; PI3K—phosphoinositide 3-kinase; Akt—protein kinase B; MAPK—mitogen-activated protein kinase; DHODH—dihydroorotate dehydrogenase; UTR—untranslated region; NSP1/NSP5—non-structural proteins of RV; mTOR—mechanistic target of rapamycin; SIRT1—sirtuin 1; Rab7—Ras-related protein Rab-7a; ac-FoxO1—acetylated Forkhead box protein O1; GTP-Rac1—guanosine triphosphate-bound Rac1 protein; FDA—Food and Drug Administration; ARE—antioxidant response element; Nrf2—nuclear factor erythroid 2–related factor 2; AGE—acute gastroenteritis.

## Data Availability

Not applicable.

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
