# Peer review of "Advances in the Diagnosis and Treatment of Rotavirus Infections: Narrative Review"

_ijms, 2025, doi:10.3390/ijms26189175_

Round 1
Reviewer 1 Report
Comments and Suggestions for Authors
Rotavirus is the leading cause of acute gastroenteritis in children under five year. The topic addressed to the new achievements on the field of diagnosis and treatment of rotavirus infections is highly interesting. However, the reviewed manuscript in present form seemed not to be valuable for scientific community. More than 40% of the cited articles are dated 2014 and earlier, only 36 (near 30%) of the cited articles are dated 2020 and later. Overall, the review lacks up-to-date information and references to the latest articles. Also, the comprehensive analysis of data provided should be conducted, which the review does not contain sufficiently in its current form. To make this manuscript acceptable for publication all chapters should be supplemented by a number of relevant recent studies and information presented in tables should be attentively discussed.
In addition, some point-by-point comments, which could help to improve the MS, are listed below:
- For some reason, the MS lacks the lines numeration common for the MDPI manuscript template. This complicates the revision process. If it was not made on purpose, it would be preferable to add the lines numeration.
- Introduction. In the second paragraph, a reference to ICTV should be provided when describing the taxonomy of the genus Rotavirus.
- Introduction. «On the basis of antibody reactivity and the sequence identity of the VP6 protein, RV A viruses, are classified into different serotypes: 42 for VP7 (type G) and 58 for VP4 (type P), which allows the use of confirmatory vaccines using the binomial genotyping system [6]». The authors mention that the classification by serotypes is based on the characteristics of the VP6 protein. However, the same sentence states that the classification is based on the properties of the VP4 and VP7 proteins. The way the information is given does not have sense and is somewhat misleading. In addition to the serotypical classification, the authors should also indicate the classification of rotavirus’ genotypes and the relationship between these two classifications.
- Section 3. Innovative diagnosis methods for RV Infections. 1. Polyacrylamide gel electrophoresis (PAGE). Nucleic acid electrophoresis in PAGE can hardly be classified as an innovative method, because it has been being known since the 1970s-1980s and is routine in almost all laboratories around the world working with nucleic acids and proteins. All references related to this subsection are dated 1980-1990s. It would be enough to briefly mention this method in section 2.2. "Laboratory diagnosis".
- Section 3. New Enzyme Immunoassay (New EIA). What is innovative about this method? How is the reaction speed achieved? Is the name "New EIA" generally accepted for this test, since, judging by the cited literature, a specific development is meant (Abbott TESTPACK ROTAVIRUS)? The authors cite articles from the 1990s. Have similar, but even more advanced, methods been developed since then? Is Abbott TESTPACK ROTAVIRUS currently used in diagnostics?
- Section 3. Figure 4. PAGE does not belong to immunological methods. It should be moved to "Traditional methods". On the contrary, «serological assays» would be more correctly attributed to «immunological methods».
- Section 4. The section 4 primarily consists of the tables, while it is preferable to discuss the information provided.
- Section 4. «However, a study by Wobudeya could not confirm the role of breast milk in reducing RV infection [67].» This article was released in 2011, are there more recent publications on this topic? These articles may be interesting for discussion: doi:10.1172/JCI168789, doi:10.1139/bcb-2024-0146, doi:10.1177/0890334420988239.
- Table 5. Also, there is no symbols decoding for A, B, C and I, II, III. The same categories “(II, B)” in different lines are explained differently: “(strong recommendation, low-quality evidence)” and “(weak recommendation, moderate-quality evidence)”. Moreover, likely, information in the column 3 does not match to the cited source “European Society for Pediatric Gastroenterology, Hepatology, and Nutrition/European Society for Pediatric Infectious Diseases Evidence-Based Guidelines for the Management of Acute Gastroenteritis in Children in Europe: Update 2014”. The information within Table 5 should be attentively checked.
- Table 7. Have any studies been conducted after 2018? These should be mentioned. For example, this research might be appropriate in this section: DOI: 10.1080/19490976.2023.2174407.
- Table 8. Have any studies been conducted after 2015? These should be mentioned. For example, this research might be appropriate in this section: DOI: 10.1016/j.mucimm.2025.01.002
- Table 9. There is more recent study about Genipin (DOI: 10.1038/s41598-020-72968-7), which is desired to be also mentioned.
- The authors have indicated, that Figures 5 and 6 were created on the base of information from [117]. However, I did not find this information in [117]. Please check and provide appropriate references.
- Subsection 4.3 is extremally short, contains only one reference and does not reflect even the main points of the theme of rotavirus prevention. This chapter deserves to be extended, highlighting the existing problems of anti-rotavirus vaccinations. Also, there are a number of publications addressed to the impact of breast feeding on the anti-rotavirus vaccination. The data are sometimes controversial; therefore, the topic is highly interesting for discussion. For example: DOI: 10.1371/journal.ppat.1009010, DOI: 10.1093/ecco-jcc/jjad005. Maybe this research could be also appropriate in this section: DOI: 10.1007/s12098-015-1854-8, DOI: 10.1038/s41467-018-07476-4, DOI: 10.3389/fimmu.2025.1572787, DOI: 10.1016/j.ebiom.2025.105850.
- I am not sure, that section 5 is needed. It is weird and seemed unnecessary. The second paragraph is almost the same as the introduction paragraph of the chapter 4.2.
There is also a couple of minor comments:
- Introduction. The sentence «All segments encode 12 viriones: 6 structural (VP1–VP7) and 6 non-nucleocapsid (NSP1–NSP6)». Probably, instead of «viriones» the authors meant «viral proteins». Also, NSPs is an abbreviation of «non-structural proteins». «Non-nucleocapsid protein» is a less common formulation and does not match the abbreviation used in the text.
- Section 4. Section 7 does not exist. Therefore, “presented in the Table 5 and sections 7.2.1-7.2.4.” should be “presented in the Table 5 and sections 4.2.1-4.2.4” and “Describe in section 7.2.1.” should be “Describe in section 4.2.1.” as well as “Describe in section 7.2.2.” should be “Describe in section 4.2.2.”
- Reference 60 format is not correct.
Author Response
Dear Reviewer,
We sincerely thank you for the time, effort, and expertise devoted to the evaluation of our manuscript. We are deeply grateful for the constructive and insightful comments provided, which we acknowledge as highly valuable for the scientific refinement and clinical relevance of our work.
In response, we have implemented all the suggested revisions, including the addition of updated references, a more detailed discussion of the data presented, and the correction of technical issues such as line numbering according to the MDPI template.
We are confident that these modifications have strengthened the manuscript, and we are truly grateful for your insightful guidance, which has been invaluable in enhancing the overall quality of our submission.
Point-by-point responses to the reviewer’s comments:
- For some reason, the MS lacks the lines numeration common for the MDPI manuscript template. This complicates the revision process. If it was not made on purpose, it would be preferable to add the lines numeration.
- The manuscript has been reformatted using the official template, which should resolve the issue.
- Introduction. In the second paragraph, a reference to ICTV should be provided when describing the taxonomy of the genus Rotavirus.
- The modification has been carried out as suggested.
- Introduction. «On the basis of antibody reactivity and the sequence identity of the VP6 protein, RV A viruses, are classified into different serotypes: 42 for VP7 (type G) and 58 for VP4 (type P), which allows the use of confirmatory vaccines using the binomial genotyping system [6]». The authors mention that the classification by serotypes is based on the characteristics of the VP6 protein. However, the same sentence states that the classification is based on the properties of the VP4 and VP7 proteins. The way the information is given does not have sense and is somewhat misleading. In addition to the serotypical classification, the authors should also indicate the classification of rotavirus’ genotypes and the relationship between these two classifications.
- The modification has been carried out as suggested.
- Section 2. Diagnostic methods for RV Infections
a. We have revised the manuscript to ensure that each table and figure is explicitly introduced and discussed in the text. In particular, we added introductory context and critical discussion to the tables on dehydration scales and probiotics to highlight their comparative utility and limitations. Additionally, in Section 2, we have included a new table comparing the Vesikari score and the modified Vesikari score system to provide a clear and comprehensive message to the reader. - Section 3. Innovative diagnosis methods for RV Infections. 1. Polyacrylamide gel electrophoresis (PAGE). Nucleic acid electrophoresis in PAGE can hardly be classified as an innovative method, because it has been being known since the 1970s-1980s and is routine in almost all laboratories around the world working with nucleic acids and proteins. All references related to this subsection are dated 1980-1990s. It would be enough to briefly mention this method in section 2.2. "Laboratory diagnosis".
- The modification has been carried out as suggested.
- Section 3. New Enzyme Immunoassay (New EIA). What is innovative about this method? How is the reaction speed achieved? Is the name "New EIA" generally accepted for this test, since, judging by the cited literature, a specific development is meant (Abbott TESTPACK ROTAVIRUS)? The authors cite articles from the 1990s. Have similar, but even more advanced, methods been developed since then? Is Abbott TESTPACK ROTAVIRUS currently used in diagnostics?
- In the revised version, we have clarified that the Abbott TESTPACK Rotavirus assay, while historically significant as an early rapid immunoassay, is no longer in current diagnostic use. Instead, we now emphasize more recent developments in rotavirus enzyme immunoassays. Specifically, we have incorporated up-to-date studies, including those describing newly developed VP6- and VP7-targeted EIAs with improved analytical sensitivity, faster reaction times, and better suitability for detecting low viral loads, particularly in vaccinated populations. These newer immunoassays reflect the current state of the field and are distinct from the earlier Abbott TESTPACK platform. Accordingly, we have removed the generic term “New EIA” and now refer to specific recently developed assays, as supported by the latest literature.
- Section 3. Figure 4. PAGE does not belong to immunological methods. It should be moved to "Traditional methods". On the contrary, «serological assays» would be more correctly attributed to «immunological methods».
- The modification has been carried out as suggested.
- Section 4. The section 4 primarily consists of the tables, while it is preferable to discuss the information provided.
- The modification has been carried out as suggested.
- Section 4. «However, a study by Wobudeya could not confirm the role of breast milk in reducing RV infection [67].» This article was released in 2011, are there more recent publications on this topic? These articles may be interesting for discussion: doi:10.1172/JCI168789, doi:10.1139/bcb-2024-0146, doi:10.1177/0890334420988239.
- The most recent studies on the indicated topic have been added.
- Table 5. Also, there is no symbols decoding for A, B, C and I, II, III. The same categories “(II, B)” in different lines are explained differently: “(strong recommendation, low-quality evidence)” and “(weak recommendation, moderate-quality evidence)”. Moreover, likely, information in the column 3 does not match to the cited source “European Society for Pediatric Gastroenterology, Hepatology, and Nutrition/European Society for Pediatric Infectious Diseases Evidence-Based Guidelines for the Management of Acute Gastroenteritis in Children in Europe: Update 2014”. The information within Table 5 should be attentively checked.
- An explanation of the coding has been provided beneath the table.
- Table 7. Have any studies been conducted after 2018? These should be mentioned. For example, this research might be appropriate in this section: DOI: 10.1080/19490976.2023.2174407.
- We have added a revised methodology for the selection of studies included in our review - Table 5 presents a summary of ESPGHAN/ESPID recommendations from 2014 on the pharmacological management of acute diarrhea in children, alongside selected original research articles published after the guideline release (between 2014-2025) (presented in the Table 5 and sections 4.2.1-4.2.4. The selection was based on an advanced PubMed and Scopus search, focusing on studies related specifically to RV-associated AGE, with review articles excluded; only studies conducted in human populations were included, except for section 4.2.5, where relevant in vitro and in vivo studies were also considered.
- The most recent studies on the indicated topic have been added.
- Table 8. Have any studies been conducted after 2015? These should be mentioned. For example, this research might be appropriate in this section: DOI: 10.1016/j.mucimm.2025.01.002
- The most recent studies on the indicated topic have been added.
- Table 9. There is a more recent study about Genipin (DOI: 10.1038/s41598-020-72968-7), which is desired to be also mentioned.
- The most recent studies on the indicated topic have been added.
- The authors have indicated that Figures 5 and 6 were created on the basis of information from [117]. However, I did not find this information in [117]. Please check and provide appropriate references.
- The most recent studies on the indicated topic have been added.
- Subsection 4.3 is extremally short, contains only one reference and does not reflect even the main points of the theme of rotavirus prevention. This chapter deserves to be extended, highlighting the existing problems of anti-rotavirus vaccinations. Also, there are a number of publications addressed to the impact of breast feeding on the anti-rotavirus vaccination. The data are sometimes controversial; therefore, the topic is highly interesting for discussion. For example: DOI: 10.1371/journal.ppat.1009010, DOI: 10.1093/ecco-jcc/jjad005. Maybe this research could be also appropriate in this section: DOI: 10.1007/s12098-015-1854-8, DOI: 10.1038/s41467-018-07476-4, DOI: 10.3389/fimmu.2025.1572787, DOI: 10.1016/j.ebiom.2025.105850.
- The topic of rotavirus infection prevention has been excluded, with the manuscript now concentrating on diagnostic approaches and therapeutic strategies.
- I am not sure, that section 5 is needed. It is weird and seemed unnecessary. The second paragraph is almost the same as the introduction paragraph of the chapter 4.2.
- Section 5 has been shortened, but we decided to retain it in order to preserve the clarity of the text.
There is also a couple of minor comments:
- Introduction. The sentence «All segments encode 12 viriones: 6 structural (VP1–VP7) and 6 non-nucleocapsid (NSP1–NSP6)». Probably, instead of «viriones» the authors meant «viral proteins». Also, NSPs is an abbreviation of «non-structural proteins». «Non-nucleocapsid protein» is a less common formulation and does not match the abbreviation used in the text.
- The modification has been carried out as suggested.
- Section 4. Section 7 does not exist. Therefore, “presented in the Table 5 and sections 7.2.1-7.2.4.” should be “presented in the Table 5 and sections 4.2.1-4.2.4” and “Describe in section 7.2.1.” should be “Describe in section 4.2.1.” as well as “Describe in section 7.2.2.” should be “Describe in section 4.2.2.”
- The formatting and numbering have been revised accordingly.
- Reference 60 format is not correct.
- The modification has been carried out as suggested.
Thank you in advance.
Reviewer 2 Report
Comments and Suggestions for Authors
This manuscript presents a well-structured and comprehensive narrative review on rotavirus infections, with a particular emphasis on recent advancements in diagnostic and therapeutic strategies. The authors provide a thorough synthesis of the current literature, effectively underscoring the persistent global burden of rotavirus despite the availability of effective vaccines. A significant strength of this work is its up-to-date discussion of novel molecular diagnostics and its extensive compilation of investigational antiviral agents. As an early draft, the manuscript possesses a strong foundation but would benefit from enhancements in methodological rigor, narrative clarity, and the integration of its visual elements to fully realize its potential as a high-impact publication.
The review adeptly covers a broad range of topics—from fundamental virology to clinical management—making it a valuable resource for both clinicians and researchers. The deliberate focus on literature published post-2014, particularly following the ESPGHAN/ESPID guidelines, ensures the content is highly relevant to contemporary practice and provides a timely update to the field.
The sections detailing emerging diagnostic technologies, such as Digital PCR (dPCR), nanopore sequencing, and next-generation enzyme immunoassays, are particularly robust and highlight the evolving landscape of rotavirus detection.
The compilation of over twenty investigational antiviral agents represents a significant effort and is a notable strength, offering a valuable resource for those interested in developmental therapeutics and future research directions.
The current abstract is a single, dense paragraph. To enhance readability and scannability, it should be restructured into distinct sections (e.g., Background, Objective, Findings, Conclusion).
The lengthy list of investigational drugs in the introduction is disruptive to the narrative flow. This detailed catalog would be more effectively presented in a dedicated table within the main body of the review.
The "Materials and Methods" section is currently brief for a scholarly review. To enhance the paper's credibility and reproducibility, the authors should adopt a more systematic approach. This should include a detailed description of the literature search strategy, specific databases used, explicit inclusion and exclusion criteria for study selection, and the time frame of the literature search. Adhering to established guidelines for systematic reviews (e.g., PRISMA) is highly recommended.
The inclusion of tables and figures is commendable. However, their integration into the text requires improvement. Each table and figure should be explicitly called out in the narrative, and the text should provide a clear explanation of its significance—not merely state its existence. For example, the valuable tables on dehydration scales and probiotics would benefit from introductory context and a subsequent critical discussion of their comparative utility and limitations.
The manuscript currently excels at summarization but would be strengthened by a more critical and analytical voice. For instance, when presenting diagnostic scales (Table 3), the authors should move beyond reporting sensitivity and specificity figures to offer a synthesized critique of their practical application, advantages, and drawbacks in various clinical settings. Providing expert insight into which agents show the most translational promise would elevate the discussion on investigational therapeutics.
The manuscript contains minor grammatical errors, typographical errors, and formatting inconsistencies (e.g., in tables and reference styling). A thorough proofreading is essential before submission. Attention should be paid to ensuring consistent formatting of spacing, bullet points, and headings throughout.
In summary, this is a promising and highly informative review that addresses a significant topic in global pediatric health. The authors have demonstrated considerable effort in compiling a vast amount of recent and relevant literature. By implementing a more systematic methodology, improving the clarity and flow of the narrative, and incorporating a stronger critical analysis, this manuscript has the strong potential to become a high-quality and impactful publication in its field.
Author Response
Dear Reviewer,
We would like to sincerely thank you for the thorough and thoughtful evaluation of our manuscript. Your insightful comments and constructive suggestions have been invaluable in guiding us to improve the scientific rigor, clarity, and overall quality of our work. By following your recommendations, we were able to refine the methodology, update the references, and strengthen the clinical relevance of the discussion.
We truly appreciate the time and expertise you dedicated to this review, and we are confident that the revisions made in response to your feedback have significantly enhanced the manuscript.
Point-by-point responses to the reviewer’s comments:
The current abstract is a single, dense paragraph. To enhance readability and scannability, it should be restructured into distinct sections (e.g., Background, Objective, Findings, Conclusion).
- Following the journal’s instructions, the abstract was kept as a single structured paragraph without section headings, while preserving the logical flow of background, objectives, findings, and conclusions.
The lengthy list of investigational drugs in the introduction is disruptive to the narrative flow. This detailed catalog would be more effectively presented in a dedicated table within the main body of the review.
- We have removed the listed medications as recommended, which has improved the clarity of the text.
The "Materials and Methods" section is currently brief for a scholarly review. To enhance the paper's credibility and reproducibility, the authors should adopt a more systematic approach. This should include a detailed description of the literature search strategy, specific databases used, explicit inclusion and exclusion criteria for study selection, and the time frame of the literature search. Adhering to established guidelines for systematic reviews (e.g., PRISMA) is highly recommended.
- We did not apply the PRISMA guidelines, as our article is a Narrative Review. However, we have revised the methodology and clarified the criteria for the literature search, which has been explicitly indicated in the manuscript. To avoid any ambiguity, we have also added the designation 'Narrative Review' to the title.
The inclusion of tables and figures is commendable. However, their integration into the text requires improvement. Each table and figure should be explicitly called out in the narrative, and the text should provide a clear explanation of its significance—not merely state its existence. For example, the valuable tables on dehydration scales and probiotics would benefit from introductory context and a subsequent critical discussion of their comparative utility and limitations.
- As recommended, we have added information regarding the scales to the text, focusing on their analysis. We have also incorporated more recent studies into the tables, in accordance with the Reviewer’s suggestion. In addition, we have provided a brief summary beneath each table wherever possible.
The manuscript currently excels at summarization but would be strengthened by a more critical and analytical voice. For instance, when presenting diagnostic scales (Table 3), the authors should move beyond reporting sensitivity and specificity figures to offer a synthesized critique of their practical application, advantages, and drawbacks in various clinical settings. Providing expert insight into which agents show the most translational promise would elevate the discussion on investigational therapeutics.
- We have added a clinical discussion of the scales to the text.
The manuscript contains minor grammatical errors, typographical errors, and formatting inconsistencies (e.g., in tables and reference styling). A thorough proofreading is essential before submission. Attention should be paid to ensuring consistent formatting of spacing, bullet points, and headings throughout.
- The revisions have been completed, and the entire manuscript has been reformatted using the official template.
Thank you in advance.
Round 2
Reviewer 1 Report
Comments and Suggestions for Authors
The authors have taken into account all the comments. The article can be accepted in its current form.
Reviewer 2 Report
Comments and Suggestions for Authors
Accept in present form